# Extracts from Wallis Sponges Inhibit *Vibrio harveyi* Biofilm Formation

**DOI:** 10.3390/microorganisms11071762

**Published:** 2023-07-06

**Authors:** Flore Caudal, Sophie Rodrigues, Alain Dufour, Sébastien Artigaud, Gwenaelle Le Blay, Sylvain Petek, Alexis Bazire

**Affiliations:** 1Laboratoire de Biotechnologie et Chimie Marines, Université Bretagne Sud, EMR CNRS 6076, IUEM, 56100 Lorient, France; flore.caudal@univ-ubs.fr (F.C.); sophie.rodrigues@univ-ubs.fr (S.R.); alain.dufour@univ-ubs.fr (A.D.); 2IRD, Univ Brest, CNRS, Ifremer, LEMAR, F-29280 Plouzane, France; sebastien.artigaud@univ-brest.fr (S.A.); gwenaelle.leblay@univ-brest.fr (G.L.B.); sylvain.petek@ird.fr (S.P.)

**Keywords:** biofilm, *Vibrio harveyi*, sponge extract, anti-biofilm activity, marine natural products

## Abstract

Pathogenic bacteria and their biofilms are involved in many human and animal diseases and are a major public health problem with, among other things, the development of antibiotic resistance. These biofilms are known to induce chronic infections for which classical treatments using antibiotic therapy are often ineffective. Sponges are sessile filter-feeding marine organisms known for their dynamic symbiotic partnerships with diverse microorganisms and their production of numerous metabolites of interest. In this study, we investigated the antibiofilm efficacy of different extracts from sponges, isolated in Wallis, without biocidal activity. Out of the 47 tested extracts, from 28 different genera, 11 showed a strong activity against *Vibrio harveyi* biofilm formation. Moreover, one of these extracts also inhibited two quorum-sensing pathways of *V. harveyi*.

## 1. Introduction

Antimicrobial resistance (AMR) is considered one of the leading public health threats, as more and more pathogenic bacteria are resistant to multiple antibiotics, thus limiting therapeutic strategies. The World Health Organization has listed this theme as a priority, as it is estimated that drug-resistant infections contributed to nearly 5 million deaths in 2019 [1].

The situation could turn out to be worse and underestimated since most studies concerning AMR are carried out on free-living cells, while bacteria are mainly organized as biofilms in their environment. Bacterial biofilms are communities of cells located at interfaces, embedded in a self-produced matrix, composed of exopolysaccharides, proteins, lipids, and extracellular DNA. The matrix contributes to protection from the environment through providing a relatively impermeable physical barrier to toxic substances, including antibiotics [2]. The multi-layer organization of cells enhances this protection, as usually only peripheral cells are exposed to external agents [3]. Dormant bacteria are also found in the deeper layers of the biofilm and are therefore less sensitive to antibiotics that generally target actively growing bacteria [4]. The overuse of antibiotics leads to an increase in their concentration in areas of human activity. Their increasing concentration is problematic because they have been shown to induce biofilm formation, leading to an adaptive response of bacteria [5,6,7] and potentially to gene transfers from animal to human pathogens [8].

Cell proximity in the structure of biofilm indeed allows two types of communication between bacteria: (i) genetical, through horizontal gene transfer, which favors the exchange of antibiotic resistance genes (ARGs), especially in seawater environments [9], and (ii) chemical, through the perception of small molecules allowing an estimation of the population density to carry out joint actions, this phenomenon being known as quorum sensing (QS) [10]. QS-controlled phenotypes include notably bacterial virulence and the regulation of biofilm formation; it is therefore an interesting therapeutic target because short-circuiting it (quorum quenching) can impair biofilm formation and/or reduce virulence mechanisms [11].

Therapeutic approaches combining conventional antibiotics and antibiofilm agents are considered promising [12,13]. Finding new molecules specifically targeting bacterial biofilm is therefore crucial, and the marine environment is a fabulous field of exploration for that. It is an extraordinary place of interactions between organisms, whether cooperative or antagonistic, and their studies allowed the discovery of various molecules with multiple applications, including antibiofilm [14,15]. Sponges, in particular, are sessile marine filter-feeding organisms colonized by a very wide variety of microorganisms [16]. There, microorganisms find ecological niches rich in nutrients and various molecules [17], and in return they provide sponges with metabolites involved in the defense against predators and pathogens [18,19] or antibiofilm molecules. Oluwabusola et al. described, for example, in 2022 [20], psammaplin A and bisaprasin, which inhibit both the production of the virulence factor elastase and biofilm formation of *Pseudomonas aeruginosa*. 

This mutual cooperation illustrates the notion of holobiont, a superorganism composed of the sponge and its microbiota [21], and in which chemical mediations allow interaction between individuals. Bacterial communities are generally resistant to antibacterials produced by the host, allowing the selection of colonizers [22]. The origin of circulating molecules is difficult to determine, since they can be produced by the sponge or by its associated microorganisms. It is therefore better to speak of the holobiont as a potential source of metabolite-controlling pathogenic bacteria. 

In the present study, we screened 47 different holobiont extracts from 28 genera of sponge from Wallis through testing their potential activity against biofilm formation by 3 pathogenic strains: *Vibrio harveyi* ORM4, *P. aeruginosa* MUC-N1, and *Tenacibaculum maritimum* DSM 17995. We further analyzed more precisely the effect of nine of these extracts against *V. harveyi* ORM4 biofilm and quorum sensing.

## 2. Results and Discussion

### 2.1. Antibiofilm Activity of Sponge Extracts

#### 2.1.1. Antibiofilm Activity Screening on Microtiter Plate

In order to find sponge extracts (SEs) with antibiofilm activity but without biocidal activity, we first used a 96-well polystyrene microtiter plate screening assay to rapidly test our collection of 47 SEs from 28 different genera. After 24 h of growth of *P. aeruginosa* MUC-N1, or 48 h for *V. Harveyi* ORM4 and *T. maritimum*, with 10 µg/mL of SE, the OD_600_ of each well were measured and compared to the SE-free control to check the absence of biocidal activity. Under these conditions, none of the 47 SEs altered the growth of the three target bacteria, suggesting the absence of biocidal activity (data not shown). Antibiogram-like filter diffusion was also used. After 24 h of growth for *P. aeruginosa* MUC-N1 and 48 h for *V. harveyi* ORM4 and *T. maritimum* DSM 17995, the three bacteria all grew without an inhibition halo around the SE-containing filters, confirming the absence of biocidal activity.

From the same 96-well microtiter plates as above, each well was emptied, rinsed, and stained with crystal violet to quantify the biofilm formed on the polystyrene after bacterial growth. The results of antibiofilm activity towards *V. harveyi* ORM4 were analyzed first. *V. harveyi* ORM4 is as pathogen of European abalone in Brittany and Normandy [23,24]. We arbitrarily distinguished four SEs groups regarding their activities against *V. harveyi* ORM4 biofilm formation (Figure 1, zones separated by dotted lines): (group 1) probiofilm SEs that caused a significant increase in biofilm, the only representative of which is E559; (group 2) neutral SEs with low activity, allowing biofilm formation between 75 and 100% compared to the SE-free controls; (group 3) moderate antibiofilm SEs, with a biofilm equivalent to between 25% and 70% of control biofilms; and finally, (group 4) strong antibiofilm SEs that allowed the inhibition of at least 75% of the biofilm, which is constituted of 11 SEs.

In addition, the genera of the sponges from which the extractions were performed are indicated in Figure 1. The 47 extracts were obtained from 28 genera, with a maximum of 4 representatives per genus (Figure 1). It is interesting to note that some of the most active extracts were obtained from specimens of the same sponge genus. The most compelling example corresponds to the extracts E581 (81.42%), E587 (84.56%), E594 (92.95%), and E595 (93.61%), all isolated from different specimens of the genus *Hyrtios* in different locations of Wallis. These four extracts are among the most active ones, with up to 93.61% inhibition of biofilm formation. Isolated molecules or extracts from species belonging to this genus and their cytotoxic, antimicrobial, antifungal, or antioxidant activities have already been well described in the literature, but to our knowledge, they were not from the same localization, nor in the same condition of extraction, and their antibiofilm activity had not been described until now. Molecules belonging to diverse chemical classes, such as alkaloids or terpenes, are produced by these organisms [25]. Although some genera are well described in the literature for their production of biochemically diverse molecules, there are also other genera, such as *Astrosclera* which produce metabolites that are not well described in the literature. Furthermore, the antibiofilm activity of this genus showed more than 75% inhibition in biofilm formation in microtiter plates.

SEs were also tested on two other pathogenic Gram-negative bacteria, *P. aeruginosa* MUC-N1, an opportunistic human pathogen, and *T. maritimum* DSM 17995, a marine pathogen found mainly in fish farming in different parts of the world [26,27]. Very few extracts showed antibiofilm activities against these two strains, and neither was significant (Figure A1 and Figure A2). Therefore, we suggest that there might be a potential specificity of SE activity limited to certain endogenous microorganisms such as *V. harveyi*.

#### 2.1.2. Antibiofilm Activity of Selected SEs against *V. harveyi* ORM4 in a Dynamic Condition of Growth

We then selected the most active extracts of the SE group 4 from seven different sponge genera (E572, E587, E588, E590, E593, E595, E616, and E621) and the unique members of the probiofilm SE group (E559) to further visualize their effects on bacterial adhesion, which is the first step of biofilm formation. Bacteria were allowed to attach for 2 h on glass slides within flow-cell chambers in the presence of the extracts at 10 µg/mL diluted in DMSO. After a 15 min rinse step with LBS medium, bacteria were observed via optical microscopy, and the percentage of surface covered by attached bacteria was compared to an SE-free control in the presence of DMSO, which is the solvent used to suspend the sponge extracts. In order to compare the results, the percentage of surface covered by the bacteria in the presence of the extract was calculated and reported as a relative percentage compared to the control (Figure 2).

Comparing the relative percentage of surface covered in the presence of the extracts to the SE-free control showed that most of the extracts induced no modification of adhesion or had only moderate pro-adhesive activity. Only the E621 extract decreased the adhesion of *V. harveyi* ORM4.

It is interesting to note here that, for half of the extracts, the percentage of covered surface is higher than the control, suggesting potentially larger biofilms after 24 h of growth.

In addition to the relative percentage of covered surface, we investigated potential changes in the organization of bacteria during this attachment phase. For this purpose, the most representative light microscopy images of the attached bacteria were analyzed and compared with each other (Figure 3).

No effect was observed using E559, E590, or E593. The other extracts led to a change in the organization of bacteria on the surface with two main phenomena: cell aggregation and/or altered adhesion. Using E587 and E595, bacteria seemed to get closer, as if they were attracted to each other, and to form small aggregates in some places. Use of E572, E588, E616, or E621 decreased bacterial adhesion to the surface. These images corroborate the results obtained in Figure 2, except for E616.

Attached bacteria were fed with a flow of medium to estimate the consequence of the sponge extracts on bacterial biofilms subsequently formed in dynamic conditions. Confocal laser scanning microscopy (CLSM) was used to observe biofilm architectures (Figure 4), and images analyses allowed us to calculate the biovolumes (µm^2^/µm^3^), the maximum thicknesses (µm), and the average thicknesses (µm) (Table 1).

When biofilms were formed in flow cell chambers under these dynamic conditions, *V. harveyi* ORM4 colonized the whole surface and formed a dense and rough biofilm with significant topographical disparities. Since the extracts were only present during the attachment step, less inhibition was expected than in the microplate assays. Despite this contact of only 2 h with the bacteria, it was possible to show a significant effect on the biofilms formed.

The addition of SE during the adhesion step primarily resulted in a decrease in the biovolume, meaning that there were fewer bacterial cells in the biofilm formed after 24 h. This effect was followed by a reduction in the mean thickness and, to a lesser extent, in the maximum thickness. The *V. harveyi* ORM4 biofilm was thus less dense; its topography remained globally the same with few decreases in maximum thickness, but we observed an area without any bacteria, like a hole in the biofilm. We thus defined three groups: strong biovolume inhibitors, between −30 and −40% (E559, E590, E616, and E621); moderate biovolume inhibitors, around −20% (E572, E587, and E595); and an extract without effect (E588).

Surprisingly, E559 was the extract with the maximum inhibitory activity, while we had chosen it as a control from the probiofilm group defined during the microplate experiment. E559 showed more than 40% inhibition in the biovolume of the biofilm, and approximately 30% inhibition in its average thickness but also in its maximum thickness. Without questioning the classical approach used by many teams, which consists of using microplates to screen for antibiofilm activity, since we observed mostly the same effects between the two experiments, we must, however, remain cautious regarding this screening. Indeed, the conditions in microplates are different at several levels compared to the experiments in flow cells. In microplates, the extracts are constantly present in the medium whereas they are evacuated by the flow in the flow cell; moreover, the culture medium is not continuously renewed in microplates and the attachment surface is also different: polystyrene for microplates and glass in the flow cell chamber.

A second example, E621, from a sponge belonging to the same genus as E559, *Petrosia*, also showed a significant decrease in *V. harveyi* ORM4 biovolume (34.65%) and average thickness (28.83%), suggesting that this genus could be a good producer of antibiofilm metabolites. Other sponge genera showed high activities in flow-cell chambers, such as *Leiosella* (E593) or *Astrosclera* (E616). The two extracts from sponges of the genus *Hyrtios*, E587 and E595, showed similar activities on the three biofilm parameters evaluated, with a more important decrease in the biovolume and then in the average thickness.

In the adhesion results presented above (Figure 2 and Figure 3), it was sometimes possible to highlight a greater surface covered by the bacteria in the presence of the extracts, suggesting that we should expect increased biofilm formation. However, as it can be seen in Table 1, all the extracts had an inhibitory effect on at least one parameter of the *V. harveyi* ORM4 biofilm, except for extract E588. This suggests that the effect of the extracts may not be due to an effect on adhesion. Since the extract was present only during the adhesion step, various scenarios can explain these biofilm impairments. The extract could have been adsorbed on some places of the surface, explaining the holes, or released during the biofilm formation phase, or it could have modified the biology or chemistry of the cell, leading to long-term effects that could have reduced biofilm production or a caused a decrease in its stability. Dheilly et al. [14] observed similar results with alterocin, an antibiofilm protein which reduced biofilm formation of *P. aeruginosa* when applied during the attachment step only [28].

### 2.2. Extracts and Quorum Sensing

Since some steps in bacterial biofilm formation are regulated via quorum sensing [29,30], we investigated whether the most active extracts influenced it. We used the *V. harveyi* BB120 bioreporter, which produces luminescence in response to QS [31]. For this purpose, *V. harveyi* BB120 was grown with or without sponge extracts at 10 µg/mL and a luminescence kinetics analysis was performed in Erlenmeyer flasks over 10 h with a sample collected every hour. The kinetics of bacterial luminescence in the presence of each extract were compared to the control.

The nine most active extracts previously studied were therefore all tested for their effects on both growth (Figure 5B) and luminescence (Figure 5A) kinetics of *V. harveyi* BB120. Seven extracts, including E590, did not show any modification of the luminescence kinetics, but two extracts, E559 and E616, modulated these kinetics in terms of the maximum luminescence expressed but also the temporality.

Using E559, peak luminescence was reached two hours earlier than the control, and with a decrease in the global luminescence of 21%. Activity was also observed for the E616 extract, with a slight delay of the luminescence peak and a decrease in its maximum luminescence.

The growth kinetics of *V. harveyi* BB120 (Figure 5B), were not altered regardless of the extract used, suggesting that luminescence modifications came from an alteration of part of the QS system.

Since the QS of *V. harveyi* is composed of three interdependent systems, Lux PQ, Lux N, and CqsS, three double mutants (JMH 597, JMH 612, and JAF 375) [32] were used to identify which system was impacted by the extracts. Each mutant showed two out of three inactivated QS pathways, so they allowed us to distinguish precisely which pathway the extracts impaired. The JAF375 mutant showed activity depending on the CqsS pathway, JMH597 depending on the LuxPQ pathway, and JMH 612 on the LuxN pathway.

As performed previously on BB120, luminescence kinetics were followed on each of the three mutants with or without E559 and E616. Unfortunately, the previously reported effects of E616 extract on BB120 were not found in these different mutants, suggesting that E616 probably had no effect on QS; or in a different manner which does not involve CqsS, LuxPQ, or LuxN; or directly on luciferase. In fact, the *V. harveyi* QS system is made up of these three interdependent pathways (LuxN, LuxPQ, and CqsS), which all converge on a cytoplasmic protein, LuxU, leading to a regulatory protein, LuxO, which in turn leads to the response regulator, LuxR [32].

However, extract 559 decreased the maximum luminescence of JMH597 (B) and JMH612 (C) mutants (Figure 6); indeed, in both cases, the effect led to a decrease in the maximum luminescence by about one third. For JAF375, maximum luminescence was not reduced, but as for the JMH597 mutant, maximum luminescence was observed earlier with a shift of nearly 4 h. The variations described in this luminescence kinetics analysis suggest that extract 559 influences all three CqsS, LuxPQ, and LuxN pathways of *V. harveryi* QS, suggesting that a common actor of the phosphorelay cascade could be impacted, such as LuxU and LuxO or small RNA [29,31,33].

Altogether, our results suggest that Wallis sponges are able to produce various metabolites that prevent the formation of *V. harveyi* biofilms, presumably to avoid colonization and subsequent disease induced by this pathogen. This discovery could lead to biotechnology applications to safeguard other marine organisms susceptible to *V. harveyi*, such as oysters or abalone, for example, through implanting sponges in the same area. This defense mechanism appears to be specific to certain species such as *V. harveyi* since the biofilm of *P. aeruginosa* and *T. maritimum*, two other pathogens, are not altered. The SEs seems to target different steps or regulators of biofilm formation, since we identified that the QS was altered for only two of the nine most active extracts. The identification, purification, and characterization of active metabolites as well as the search for the exact target remain to be performed.

## 3. Materials and Methods

### 3.1. Biological Material

Sponges were collected by hand using SCUBA in Wallis and Futuna during the sampling cruise Wallis 2018 aboard the R/V Alis, off the coast of Wallis Island (18–31 July 2018, 13°16.367′ S; 176°12.283′ W) between 6 and 50 m deep [34]. Voucher samples were deposited at the Queensland Museum (Brisbane, Australia) and identified by Dr. Merrick Ekins; accession numbers available in the Appendix A. Sponges were deep-frozen onboard until work. They were then freeze-dried, grounded, and extracted.

*V. harveyi* ORM4 is a pathogen of the European abalone *Haliotis tuberculata*, a marine gastropod. This bacterium was isolated during a severe epidemic resulting in the mortality of more than half of the natural stock of European abalone in Brittany and Normandy (France) in the late 1990s [23,24]. Unless otherwise stated, *V. harveyi* ORM4 was routinely grown at 20 °C in saline Luria-Bertani (SLB) medium (NaCl 20 g/L, tryptone 10 g/L and yeast extract 5 g/L) under orbital shaking (125 rpm).

*V. harveyi* BB120 [31] wild type and its derived double mutants, *V. harveyi* JAF 375, JMH 597, and JMH 912 [32,35], were used as models for quorum sensing studies. They were cultivated at 28 °C in autoinducer bioassay (AB) medium (0.3 M NaCl, 0.05 M MgSO_4_, 0.2% vitamin-free casamino acids (Difco) adjusted to pH 7.5 and sterilized via autoclaving; to 970 mL of this mix, 10 mL of sterile 1 M potassium phosphate (pH 7.0), 10 mL of 0.1 M L-arginine (free-base), and 20 mL of sterile 50% glycerol was added) [36].

*P. aeruginosa* MUC-N1 is a mucoid clinical isolated from a cystic fibrosis patient at the Nantes teaching hospital [28,37]. *P. aeruginosa* MUC N1 was grown at 37 °C in Luria-Bertani medium (NaCl 10 g/L, tryptone 10 g/L, and yeast extract 5 g/L) under orbital shaking (125 rpm).

*T. maritimum* DSM 17995, a marine pathogenic bacterium, was grown at 28 °C in Marine Broth (Difco) under orbital shaking (125 rpm) [38].

### 3.2. Sponge Extraction

The lyophilized samples were extracted using the following pressurized liquid extraction process. PLE experiments were performed using an accelerated solvent extractor system (ASE 150, Dionex Corporation, Sunnyvale, CA, USA). The following solvents were used: methylene chloride, ethanol, and methanol (Sigma-Aldrich, Saint-Quentin Fallavier, France).

A mixture of 8 g of freeze-dried powder of sponge and 8 g of sand of Fontainebleau was poured into a 34 mL stainless steel extraction cell, fitted with glass fiber filters at its ends. The extraction was performed with a mixture of CH_2_Cl_2_/MeOH (50/50) at 45 °C under 100 bar during 5 static cycles of 5 min. A second extraction was performed with the same parameters after the cell was turned upside down, rinsed with solvents, and flushed with nitrogen. The combined organic extracts were filtered on glass wool before being concentrated under reduced pressure. The dry extract was desalted using absolute EtOH via resuspension, centrifugation, and collection of the supernatant, 3 times successively. The combined supernatants were concentrated under reduced pressure to provide the dry crude extract which was finally stored at 4 °C, prior to analysis and bioassay.

### 3.3. Antibiofilm Assays

#### 3.3.1. Microtiter Plate Assay (Static Conditions/Polystyrene Surface)

*V. harveyi* ORM4 biofilms were formed at 20 °C under static conditions in sterile 96-well microtiter plates (Thermo Scientific™ Nunc™ MicroWell™ 96-Well) for 48 h as described in the protocol of Coffey and Anderson [39]. Briefly, 200 µL of *V. harveyi* ORM4 suspension, diluted to an OD_600_ of 0.1 in LBS medium, were inoculated in each well except for the blank, where non-inoculated LBS medium was used. Peripheral wells contained 200 µL of autoclaved distilled water to reduce the risk of evaporation. The different extracts were added to the 200 µL of *Vibrio* suspension to a final concentration of 10 µg/mL (1% DMSO), and DMSO was used as a control. For each condition, technical duplicates and biological triplicates were performed.

After 48 h of growth, OD_600_ of each well was measured. Staining and quantification of biofilms were performed according to Coffey and Anderson’s protocol. Finally, the absorbance of each well was measured to quantify biofilm formation under static conditions.

#### 3.3.2. Anti-Adhesion Assay (Static Conditions/Glass Surface)

Sponge extract’s effects on the attachment of bacteria were evaluated using a three-channel flow cell chamber (channel dimensions 1 by 4 by 40 mm, Technical University of Denmark Systems Biology, Lyngby, Denmark). The flow cell system was assembled, prepared, and sterilized as described by Tolker-Nielsen and Stenberg [40,41]. Each channel was inoculated with 300 µL of an overnight *V. harveyi* ORM4 culture diluted to an OD_600_ of 0.1 in sterile ASW (Artificial Sea Water) to which was added either 1% DMSO (SE-free control) or an extract diluted in DMSO at a final concentration of 10 µg/mL. Bacteria were allowed to attach for 2 h to the glass surface. After 10 min of rinsing, adhered cells were observed in light transmission with the CLSM. Bacterial adhesion was evaluated via determining the percentage of glass covered by cells through image analysis using ImageJ. For each tested condition, at least four images of three independent experiments were analyzed.

#### 3.3.3. Impact on Biofilm Formation in Flow Cell Chamber Assay (Dynamic Conditions/Glass Surface)

After the attachment step, biofilms were grown under hydrodynamic conditions with a constant flow (2.5 mL/h) of LBS medium SE-free for 24 h at 20 °C. Biofilms were then observed via CLSM as described below.

#### 3.3.4. Confocal Laser Scanning Microscopy (CLSM)

Biofilms formed by *V. harveyi* ORM4 were stained through injecting each channel with 300 µL of 5 µM of SYTO^TM^ 9 Green (Invitrogen, Carlslab, CA, USA) prepared in sterile ASW, without flow, at room temperature, in the dark for 15 min. A 10 min wash with a flow of medium at 2.5 mL/h was then performed, which was immediately followed by CLSM observations performed with an LSM 710 microscope (Zeiss, Oberkochen, Germany) using a 40× oil immersion objective. SYTO 9 dye was excited at 488 nm and fluorescence emission was detected between 500 and 550 nm. For the three-dimensional (3D) visualization, images were taken every micrometer throughout the whole biofilm depth. The biofilm stacks were analyzed using COMSTAT v.1 2000 software [42] to estimate the biovolume (µm^3^_/_µm^2^), average thickness (µm), and maximum thickness (µm). For each tested condition, at least three image stacks of three independent experiments were analyzed.

### 3.4. Antibacterial Assays

To assess the antibacterial activity of the different extracts, filter diffusion assays were realized [43]. Solid LBS plates were inoculated with overnight bacteria preculture and 5 µL of extract (10 µg/mL). Plates were then incubated at 20 °C for marine isolates and 37 °C for *P. aeruginosa* during 48 h or 24 h, respectively. The growth inhibition zones were then measured.

### 3.5. Anti-Quorum Sensing Assays

*V. harveyi* BB120 was grown overnight in Zobell medium (yeast extract 1 g/L, tryptone 4 g/L and sea salts 30 g/L), diluted to an OD_600_ of 0.1 in AB medium for the bioluminescence kinetic. Mutants were grown under same conditions with kanamycin 100 mg/L for JAF 375 and chloramphenicol 10 mg/L for JMH 597 and JMH 612. Growth and bioluminescence were monitored every hour with an infinite M200 PRO plate reader (Tecan, Männedorf, Switzerland). For each extract, luminescence kinetics were performed in triplicates.

Bioluminescence kinetics analysis was performed in 96-well plates (Thermo Scientific^TM^ Flat Bottom White Plates, Waltham, MA, USA) using an infinite M200 PRO plate reader (Tecan, Männedorf, Switzerland). Bioluminescence was expressed in relative light units (RLU).

### 3.6. Statistical Analysis

All experiments were conducted in triplicate and the software R-4.2.2 was used for statistical analysis using the Kruskal test followed by a Dunn test. A *p* value of <0.05 was taken as significant.

## 4. Conclusions

Based on a screening of 47 sponge extracts isolated in Wallis, this study demonstrated the very strong antibiofilm activity of 9 of these extracts against the marine pathogen *V. harveyi* ORM4. The effects of these different extracts on biofilm architecture were observed via confocal laser scanning microscopy. In addition, these extracts had no biocidal activity against the target pathogen, limiting the associated risks of resistance. The next steps will be to identify, purify, and characterize the active metabolites and then find the exact target. This study is one of the first to report on the antibiofilm activity of sponge extracts isolated from Wallis.

## Figures and Tables

**Figure 1 microorganisms-11-01762-f001:**
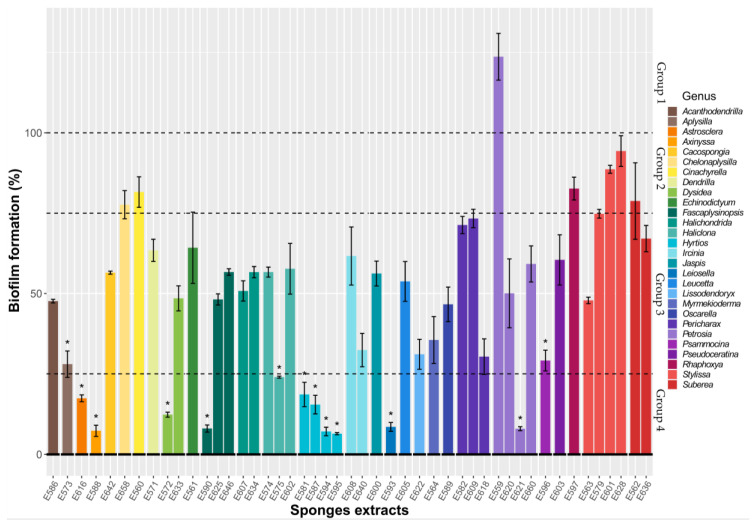
Inhibition of *V. harveyi* ORM4 biofilm formation by different sponge extracts in microtiter plates. *V. harveyi* ORM4 was inoculated with SE (10 µg/mL) in 96-well microplates and incubated at 20 °C for 48 h to form biofilms. The dotted lines delimitate four groups of SEs, which have been distinguished according to their activity levels. The sponge genera are indicated. Bars represent means ± standard error of the mean for three replicates and the 100% dotted line corresponds to the DMSO SE-free control. * *p* < 0.05.

**Figure 2 microorganisms-11-01762-f002:**
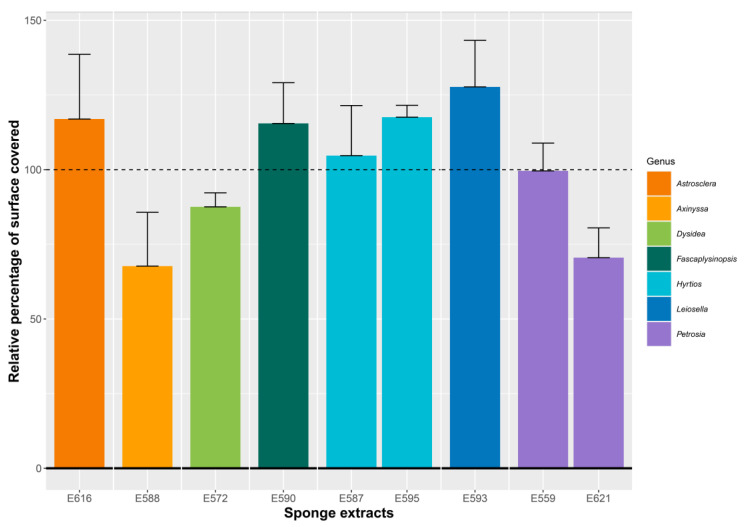
Percentage of glass surface covered by bacteria in the presence of sponge extracts compared to the SE-free control. *V. harveyi* ORM4 was inoculated with sponge extracts (10 µg/mL) in flow-cell chambers and bacteria were allowed to attach at 20 °C for 2 h. The sponge genera are indicated. Bars represent means ± standard error of the mean for three replicates and the dotted line corresponds to the DMSO control.

**Figure 3 microorganisms-11-01762-f003:**
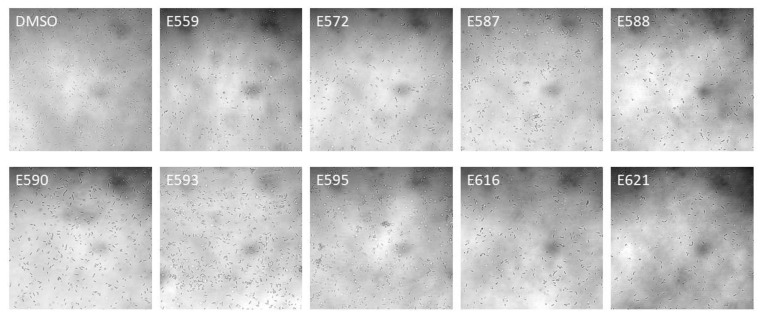
Light microscopic observation of the effect of nine sponge extracts on the adhesion of *V. harveyi* ORM4. *V. harveyi* ORM4 was inoculated with sponge extracts (10 µg/mL) in flow-cell chambers and allowed to attach on a glass surface at 20 °C for 2 h prior to observe them at 40× magnification. Images shown are representative of four images per condition from three independent experiments.

**Figure 4 microorganisms-11-01762-f004:**
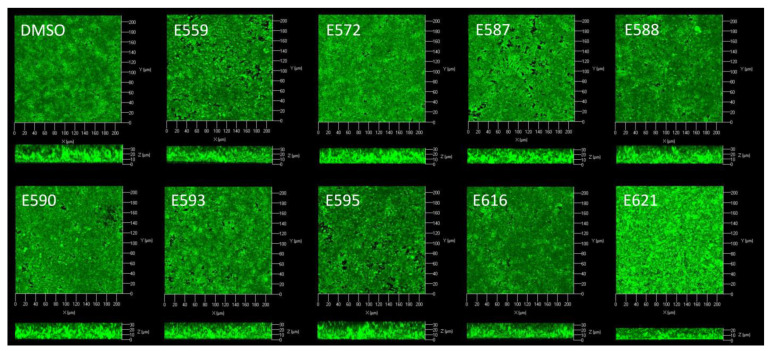
*V. harveyi* ORM4 biofilms formed in flow-cell chambers after incubation with different sponge extracts during the attachment step. *V. harveyi* ORM4 was inoculated with SEs (10 µg/mL) in flow cell chambers and biofilms were then grown at 20 °C for 24 h. Confocal laser microscopy observations with or without the addition of extracts are in 3D: top and side views are shown here.

**Figure 5 microorganisms-11-01762-f005:**
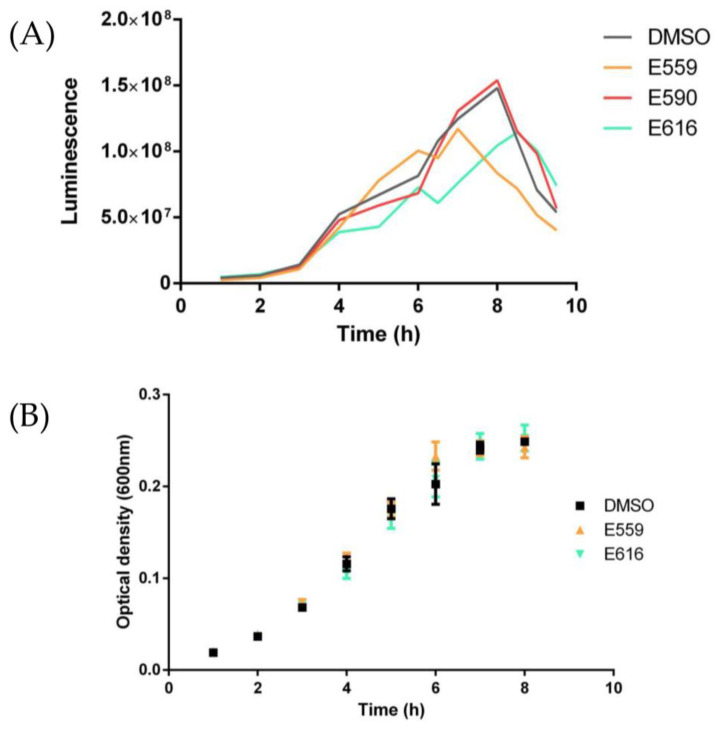
Effect of sponge extracts on *V. harveyi* BB120 quorum sensing. (**A**) Luminescence of *V. harveyi* BB120 in the presence of 10 µg/mL extract or with DMSO as control. Luminescence is counted in relative light units. (**B**) Kinetic growth of *V. harveyi* BB120 in the presence of 10 µg/mL extract or SE-free DMSO as control; growth is shown as optical density at 600 nm.

**Figure 6 microorganisms-11-01762-f006:**
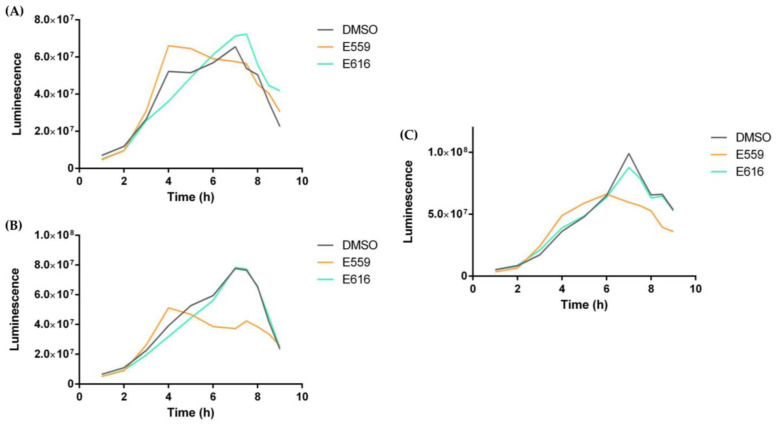
Effect of sponge extracts on *V. harveyi* JAF375 (**A**), JMH597 (**B**), and JMH612 (**C**) quorum sensing. Luminescence of *V. harveyi* JAF375 (**A**), JMH597 (**B**), and JMH612 (**C**) in the presence of 10 µg/mL extract or with DMSO as control. Luminescence is shown as RLU.

**Table 1 microorganisms-11-01762-t001:** Comparison of percentage of inhibition in biovolume and average and maximum thickness of *V. harveyi* ORM4 biofilms in the presence of sponge extracts. The results are presented as the average inhibition of three replicates ± standard error of the mean.

	E559	E590	E616	E621	E572	E587	E593	E595	E588
% Inhibition in biovolume	41.76 ± 10.01	33.20 ± 6.61	37.91 ± 3.87	34.65 ± 6.32	26.41 ± 16.01	30.35 ± 12.90	28.21 ± 14.39	26.86 ± 7.75	0.72 ± 17.41
% Inhibition in average thickness	30.59 ± 6.37	21.38 ± 4.53	13.49 ± 5.17	28.83 ± 6.86	16.85 ± 12.97	17.49 ± 16.37	15.31 ± 10.42	21.71 ± 5.71	1.59 ± 8.78
% Inhibition in maximum thickness	28.75 ± 0.97	15.18 ± 3.21	8.71 ± 5.56	11.81 ± 6.50	5.54 ± 6.19	6.35 ± 7.87	8.68 ± 5.31	19.94 ± 5.33	0.51 ± 3.45
	Strong inhibitor group				Moderate inhibitor group				No effect

## Data Availability

The data presented in this study are available on request from the corresponding author.

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
