# Peer review of "Extracts from Wallis Sponges Inhibit Vibrio harveyi Biofilm Formation"

_microorganisms, 2023, doi:10.3390/microorganisms11071762_

Round 1

Reviewer 1 Report

This paper aims to investigate the antibiofilm efficacy of different extracts from sponges, isolated in Wallis, without biocidal activity.

 This study provides the results of a screening of 47 different holobionthe extracts from 28 genera of sponge from Wallis by testing their potential activity against the biofilm formation by 3  pathogenic strains: Vibrio harveyi ORM4, P. aeruginosa MUC-N1 and Tenacibaculum mariti- mum DSM 17995 and the analysis of  more precisely the effect of 9 of these extracts against V. harveyi ORM4 biofilm and quorum-sensing.

The manuscript is written comprehensively enough to be understandable; the paper stated the purpose, discussion and global implication are clearly stated and consistent with the rest of the manuscript; authors provided the required tests and analysis and enough information in their discussion by using a good number of important articles talked about the subject.

The authors addressed their hypothesis and opinion in a reproducible way and proved their results through all the required experiments and analysis and they used enough number of analyses to prove their results. The results were presented in a clear way; however, they did not reach to a conclusion which is an important part of this research article.

The abbreviations were explained at the first place they are mentioned.

In vitro, in vivo, et al.: should be written in italic.

No plagiarism has been detected.

References: The authors did not follow the journal guidelines for some references.

Year: should be in Bold, Volume: should be in italic

i.e.: Ref 39, 42

This paper aims to investigate the antibiofilm efficacy of different extracts from sponges, isolated in Wallis, without biocidal activity.

 This study provides the results of a screening of 47 different holobionthe extracts from 28 genera of sponge from Wallis by testing their potential activity against the biofilm formation by 3  pathogenic strains: Vibrio harveyi ORM4, P. aeruginosa MUC-N1 and Tenacibaculum mariti- mum DSM 17995 and the analysis of  more precisely the effect of 9 of these extracts against V. harveyi ORM4 biofilm and quorum-sensing.

The manuscript is written comprehensively enough to be understandable; the paper stated the purpose, discussion and global implication are clearly stated and consistent with the rest of the manuscript; authors provided the required tests and analysis and enough information in their discussion by using a good number of important articles talked about the subject.

The authors addressed their hypothesis and opinion in a reproducible way and proved their results through all the required experiments and analysis and they used enough number of analyses to prove their results. The results were presented in a clear way; however, they did not reach to a conclusion which is an important part of this research article.

The abbreviations were explained at the first place they are mentioned.

In vitro, in vivo, et al.: should be written in italic.

No plagiarism has been detected.

References: The authors did not follow the journal guidelines for some references.

Year: should be in Bold, Volume: should be in italic

i.e.: Ref 39, 42

Author Response

Dear reporter, thank you for your comments, we have answered to all of them in the manuscript and highlighted the modifications in yellow. We also answer more precisely to each point below, in italic.

 This paper aims to investigate the antibiofilm efficacy of different extracts from sponges, isolated in Wallis, without biocidal activity.

This study provides the results of a screening of 47 different holobionthe extracts from 28 genera of sponge from Wallis by testing their potential activity against the biofilm formation by 3 pathogenic strains: Vibrio harveyi ORM4, P. aeruginosa MUC-N1 and Tenacibaculum maritimum DSM 17995 and the analysis of more precisely the effect of 9 of these extracts against V. harveyi ORM4 biofilm and quorum-sensing.

The manuscript is written comprehensively enough to be understandable; the paper stated the purpose, discussion and global implication are clearly stated and consistent with the rest of the manuscript; authors provided the required tests and analysis and enough information in their discussion by using a good number of important articles talked about the subject.

The authors addressed their hypothesis and opinion in a reproducible way and proved their results through all the required experiments and analysis and they used enough number of analyses to prove their results. The results were presented in a clear way; however, they did not reach to a conclusion which is an important part of this research article.

Thank you for your comments. We apologize for the unfortunate omission of a conclusion section. We have added on line 406:

“Based on a screening of 47 sponge extracts isolated in Wallis, this study demonstrated the very strong antibiofilm activity of 9 of these extracts against the marine pathogen V. harveyi ORM4. The effects of these different extracts on the biofilm architecture were observed by confocal laser scanning microscopy. In addition, these extracts had no biocidal activity against the target pathogen, limiting the associated risks of resistance. The next steps will be to identify, purify and characterize the active metabolites and then find the exact target. This study is one of the first to report on antibiofilm activity of sponge extracts isolated from Wallis.”

The abbreviations were explained at the first place they are mentioned.

In vitro, in vivo, et al.: should be written in italic.

Of course, you are right, we modify this everywhere, it’s underlined in yellow.

No plagiarism has been detected.

References: The authors did not follow the journal guidelines for some references.

Year: should be in Bold, Volume: should be in italic (i.e.: Ref 39, 42)

Thank you for this precision, all references have been corrected.

Reviewer 2 Report

The manuscript entitled “Extracts from Wallis sponges inhibit Vibrio harveyi biofilm formation.” is about to provide the Wallis sponges extract as an anti-biofilm formation candidate.

It is interesting that authors said that SE did not inhibit the bacterial growth but 4 extracts showed antibacterial activity at line 109. Authors need to explain that at discussion.

It is not clear what is the “glass surface covered” at line 153? That is not looks like counting cells. Authors need to clarify that and also how to check the coverage.

It is curious why don’t authors use cell chamber not microplate to check the biofilm formation because the difference of material should change or influence the biovolume formation, and also made in-consistent results compared with anti-biofilm formation.

Authors need to make sure how to treat the extracts, because extract should be treated to bacteria for 2 hr that should be influenced to the adherence in the results section, but it looks like for 24hr in the material and method section. Also, based on that authors need to make sure the extracts influence the adherence or biofilm formation. It is important because even though the adherence is the first step of biofilm formation, both of them is not same, so authors need to make sure of them.

What is the “dynamic conditions” at line 184?

Authors need to increase the figure resolution

It is curious whether the inhibition of QS by treatment of sponge extract influence the bacterial biofilm formation or not, because QS just inhibit the QS system by only 20% but biofilm inhibition is grater than that. So, authors need to explain what make this difference with ratio of inhibition.

Authors need to use abbreviated scientific name after first time use.

Authors need to provide the references for the “3.4. antibacterial assays”.

Author Response

Dear reporter, thank you for your comments, we have answered to all of them in the manuscript and highlighted the modifications in yellow. We also answer more precisely to each point below, in italic.

The manuscript entitled “Extracts from Wallis sponges inhibit Vibrio harveyi biofilm formation.” is about to provide the Wallis sponges extract as an anti-biofilm formation candidate.

It is interesting that authors said that SE did not inhibit the bacterial growth but 4 extracts showed antibacterial activity at line 109. Authors need to explain that at discussion.

That's a good point indeed, we misspoke. So, we've amended the text to specify that molecules and extracts from species in these genera have already been described, notably for their antimicrobial activity. But not in our species, and not under the same extraction conditions.

It is not clear what is the “glass surface covered” at line 153? That is not looks like counting cells. Authors need to clarify that and also how to check the coverage.

You are right, Material and methods are not sufficiently clear, thank you for your comment. We analyzed the percentage coverage of glass surfaces in the flow-cell using ImageJ software. For each tested condition, at least four images of three independent experiments were analyzed. We've modified the material and method to make it clearer and hopefully more understandable.

It is curious why don’t authors use cell chamber not microplate to check the biofilm formation because the difference of material should change or influence the biovolume formation, and also made in-consistent results compared with anti-biofilm formation.

That's an excellent point, perhaps our words aren't clear enough to explain the approach we used. Screening can't be carried out on plastic flow cells because the microscope's lasers can't pass through them efficiently, and microplates fully made of glass aren't generally used for screening. This is the approach used in the literature for this type of experiment, but no one had described this bias before. We thought it was important to show these results.

Authors need to make sure how to treat the extracts, because extract should be treated to bacteria for 2 hr that should be influenced to the adherence in the results section, but it looks like for 24hr in the material and method section. Also, based on that authors need to make sure the extracts influence the adherence or biofilm formation. It is important because even though the adherence is the first step of biofilm formation, both of them is not same, so authors need to make sure of them.

Thank you for this comment, on rereading the materials and methods, we do indeed realize the misunderstanding. In this experiment, we only applied extracts during the bacterial attachment step. There is no extract in the culture medium feeding the biofilm afterwards, because we don't have enough extract available. We have already carried out experiments under such conditions (Dheilly et al. 2010) and indeed there may or may not be differences in adhesion, and subsequent differences in the biofilm. We have modified the materials and methods section line 360 to specify that the attachment step is carried out in a flow chamber without flow for 2h, in the presence or absence of extracts, and that the flow was then applied for the subsequent step of biofilm formation with flow but without extracts.

What is the “dynamic conditions” at line 184?

We probably were no clear enough. Dynamic conditions correspond to flow-cell assays, while static conditions correspond to microplate assays. In flow-cell chambers, a constant supply of nutrients and oxygen is provided at a flow rate of 2.5 ml/h, whereas in microplates the culture medium is not renewed during the all experiment. We change line 186 (184 in the previous manuscript) by When biofilm was formed in flow cell chamber, under dynamic conditions. It’s underlined in yellow in the text.

Authors need to increase the figure resolution.

Of course, we can. It’s done.

It is curious whether the inhibition of QS by treatment of sponge extract influence the bacterial biofilm formation or not, because QS just inhibit the QS system by only 20% but biofilm inhibition is greater than that. So, authors need to explain what make this difference with ratio of inhibition.

It's not so strange: QS is involved in biofilm structuring, but it's not the only phenomenon involved. So, there's no proportionality between biofilm inhibition and QS inhibition, it's just an element of understanding.

Authors need to use abbreviated scientific name after first time use.

Sorry for the oversight, it’s corrected.

Authors need to provide the references for the “3.4. antibacterial assays”.

We added the ref [43]
